# Rare Benign Tumors and Tumor-like Lesions of the Hand without Skin Damage—Clinical, Imagistic and Histopathological Diagnosis, Retrospective Study

**DOI:** 10.3390/diagnostics13061204

**Published:** 2023-03-22

**Authors:** Mihaela Pertea, Oxana Madalina Grosu, Alexandru Filip, Dan Cristian Moraru, Stefana Luca, Madalina-Cristina Fotea, Sorinel Lunca, Doinita Olinici, Vladimir Poroch, Claudiu Carp, Bogdan Veliceasa

**Affiliations:** 1Department Plastic Surgery and Reconstructive, Faculty of Medicine,“Grigore T. Popa” University of Medicine and Pharmacy, 700115 Iasi, Romania; 2Department of Plastic Surgery and Reconstructive Microsurgery, “Sf. Spiridon” Emergency County Hospital, 700111 Iasi, Romania; 3Department of Orthopaedics and Traumatology, “Sf. Spiridon” Emergency County Hospital, 700111 Iasi, Romania; 4Second Surgical Clinic, Regional Institute of Oncology, 700483 Iasi, Romania; 5Department of Dermatology, “Sf. Spiridon” Emergency County Hospital, 700111 Iasi, Romania; 6Department of Palliative Care, Regional Institute of Oncology, 700483 Iasi, Romania

**Keywords:** hand, fingers, rare tumor, rare tumor-like lesion, surgery

## Abstract

Background: The broad spectrum of diagnoses and clinical features of hand tumors and the absence of pathognomonic signs often lead to an inaccurate or delayed diagnosis. However, only a few reports have comprehensively referenced the diagnosis and clinical features of hand tumors. This study intends to highlight the clinical, imaging and histological characteristics of uncommon hand tumors or tumor-like lesions. Methods: In this retrospective study, we report a series of 80 patients diagnosed with rare hand tumors and tumor-like lesions without skin damage. Age, gender, tumor location, imaging examinations and clinical and laboratory findings were analyzed. The histopathological exam established the final diagnosis. Surgery was indicated and performed in all cases. Results: This study included: neurofibroma, glomus tumor, lipoma, schwannoma, epidermal inclusion cyst and idiopathic tenosynovitis with “rice bodies.” We have described the clinical, imagistic and histopathological particularities of these tumors. Surgical management included the complete removal of tumors, with no recurrence recorded within two years and overall high patient satisfaction. The most common findings were lipomas and the rarest neurofibromas. Conclusions: To optimize the care of hand tumors and reduce diagnostic and treatment errors, knowledge of hand tumor types and their clinical and laboratory characteristics is necessary for every surgeon.

## 1. Introduction

The hands, a symbol of action and activity, are highly sophisticated and specialized body parts. The hand receives, holds, gives, expresses communion and prays. Both hands represent only 2% of the total body surface area and only 1.2% of the total body weight [1]. However, hand tumors account for 15% of all soft tissue tumors [1,2]. Moreover, 95% of the soft tissue tumors on the hand are benign and tumor-like lesions [2,3]. However, benign tumors and tumor-like lesions of the hand have a low incidence compared to other anatomical sites, some of them being included in the category of rare tumors; therefore, diagnosis and surgical treatment require good knowledge and skills [3,4]. A delayed diagnosis often results in delayed and consequently difficult treatment, both for the patient and the surgeon [1,5]. With tumor growth, there is significant tissue destruction that often requires a more complex surgical approach [1,3]. This often results in increased morbidity, functional and quality-of-life impairment, or delayed recovery [1,6]. Surgical interventions must restore hand function as close to normal as possible in all cases, with the best possible esthetic result [2,3,7]. Only a few reports have comprehensively referenced the clinical features and diagnosis of benign tumors or tumor-like lesions of the hands, such as lipomas, schwannomas, neurofibromas, glomus tumors, tenosynovitis with rice bodies, or epidermal inclusion cysts on the palm [2,3,8,9,10,11,12]. We retrospectively reviewed the distribution of pathological diagnoses, the clinical, imaging, anatomopathological and, when appropriate, immunohistochemical features of 80 patients diagnosed with hand tumors who underwent surgical treatment at our institute. The study included a group of 80 patients with hand palm and finger lipomas, neurofibromas located on the nail bed, glomus tumors of the fingers, schwannomas of the palm and fingers, as well as tumor-like lesions of the idiopathic tenosynovitis with rice bodies type and large epidermal inclusion cysts located on the palm.

## 2. Materials and Methods

The 80 patients diagnosed with tumors or tumor-like lesions of the hand included in this study had been admitted to and treated in the Iasi “Sf Spiridon” Hospital between 2014 and 2020. All patients were informed and signed informed consent, and the Hospital Ethics Committee approval no 157/22.11.2022 was obtained. Inclusion criteria in the study were related to demographic characteristics (age ≥ 18 years of age, male or female gender) and histopathological characteristics (benign or soft tissue tumor-like lesion). None of the types of soft tissue tumors included in the study had skin involvement. Exclusion criteria were related to demographic characteristics (age < 18 years of age, male or female gender) and clinical and histopathological characteristics (bone tumors, malign lesions). All 80 patients were subjected to the analysis of medical history, revealing the time interval from tumor occurrence to the first visit to the doctor, the symptoms and clinical signs specific to each type of tumor, as well as imaging, histopathologic and immunohistochemistry characteristics. In all cases, the imaging investigations consisted of a two-view of the radiological exam (Rx), ultrasound exam (USG) and in cases with an uncertain diagnosis, magnetic imaging resonance exam (MRI). In all cases, surgical treatment was performed under wide-awake local anesthesia with no tourniquet (Walant) or loco-regional anesthesia (axillary or infraclavicular block, in the latter case, using a tourniquet-type hemostatic tourniquet applied around the arm). In all cases, an excisional biopsy was performed instead of an initial biopsy followed by excision. In cases of suspected schwannoma, punch biopsy was avoided to limit nerve trauma due to the nervous nature of the lesion. Tumors larger than 5 cm required laborious surgical interventions, with the postoperative indication of functional sequels, physical therapy, or occupational therapy for functional re-education. The immediate postoperative results concerning the neurological symptoms were evaluated using the British Medical Research Council (BMRC) scale modified by Omer and Dellon. The assessment of outcomes was performed by sensitive and motor (active and passive) evaluation and confirmed by specific tests, such as two-point discrimination (2PD) and Semmes–Weinstein (SW) test, and the calculation of the disabilities of the arm, shoulder and hand (DASH) score. The degree of postoperative satisfaction of the patients was evaluated using the Michigan hand outcomes questionnaire (MHQ) scale, which includes six criteria: overall hand function, daily living activities, pain intensity, work activities, aesthetic aspect and patient satisfaction. All patients were subjected to the analysis of age, gender, tumor occurrence, progression and size, particularities revealed by clinical and imaging examination (Rx, USG, MRI), surgical treatment and outcome, as well as histopathological and immunohistochemical findings.

## 3. Results

Of the 80 patients included in the study, 50 were female (62.5%) and 30 were male (37.5%). The following diagnoses were made: neurofibroma (2 cases/2.5%), glomus tumor (15 cases/18.75%), lipoma (30 cases/37.5%), schwannoma (22 cases/27.5%), epidermal inclusion cyst (3 cases/3.75%) and idiopathic tenosynovitis with rice bodies (8 cases/10%) (Figure 1).

The epidemiological data and tumor history are listed in Table 1.

Clinical and macroscopic aspects of the tumors in the study group are listed in Table 2. 

The findings of the imaging investigations of the tumors in the study group are listed in Table 3. 

Histopathological and immunohistochemical features of the tumors in the study group are listed in Table 4.

### 3.1. Neurofibroma

First described by Von Recklinghausen, neurofibromas are benign peripheral nerve sheath tumors most commonly associated with neurofibromatosis [10]. Common sites include fingers and toes and typically develop asymptomatically as slowly enlarging soft lesions in females and in the second or third decade of life [13,14,15]. Subungual localization is very rare, with only 11 cases reported in the literature, most often with nail deformation [10]. In the absence of noisy symptomatology or pathognomonic signs, this rare type of tumor located in the nail bed is difficult to diagnose; thus, creating a differential diagnosis is essential (glomus tumors, subungual hemangiomas). Glomus tumors can be excluded in the absence of pain. A plain, two-view X-ray may reveal a possible bone imprinting (distal phalanx) and exclude bone tumors’ presence. Nail bed neurofibroma was diagnosed in 2 of the 80 studied cases.

In both cases, the patients were female, with nail bed involvement of the second digit (D2) in one case and the volar aspect of the second phalanx of the fifth digit (P2D5) in the other case (Figure 2).

The two patients were 42 and 51 years old, respectively. The tumor took a progression of 2 years and 3.5 years, respectively. In both cases, the patients did not report any clinical signs. Plain X-ray and ultrasonography were performed, revealing the presence of a hyperechoic nodular tumor with a polycyclic profile and low Doppler signal. In one of the cases, bone imprinting at the level of the distal phalanx was observed. No biopsy was performed. In both cases, surgery was performed under the Walant technique. After the surgical removal of the tumor, the excised piece was examined histopathologically and immunohistochemically (Figure 3).

Immunohistochemical staining for S100 protein (Figure 4) and CD34 was positive. The differential diagnoses considered lipoma, schwannoma or giant cell tumor of synovial staining after the sheath. Tumor consistency and mobility on the deep planes were not suggestive of bone tumor; radiologic findings excluded this diagnosis. The immediate and long-term postoperative results were among the best, with the resumption of total activity and satisfactory aesthetic appearance in the case of nail bed neurofibroma. No recurrences were recorded within two years.

### 3.2. Glomus Tumors

Glomus tumors are uncommon benign tumors involving the glomus body, an apparatus involved in the thermoregulation of cutaneous microvascularization. Location on the volar aspect of the hand is reported in only 10% of all cases with glomus tumor of the hand [16]. Glomus tumors present as a classic triad of severe pain, point tenderness, and cold sensitivity and have a relatively short history of disease. In the study group, we recorded 13 female patients and 2 males with a history of disease of 6 months to 3 years. High-intensity pain to the touch and the positive clinical tests triad (Love’s test, cold sensitivity test, Hildreth’s test) were recorded in all cases. The imaging examination included face and profile X-rays and ultrasound (in 10 of the 15 cases).

The remaining five patients could not tolerate being touched with the ultrasound probe. MRI was not performed in any of the cases for economic reasons. Surgery was indicated in all cases and was performed under Walant anesthesia. For lesions with nail bed involvement, surgical management involved complete tumor ablation by initially removing a portion of the nail blade/plate, nail bed incision, nail bed suturing, and nail plate repositioning for protection. In all the other cases, complete tumor ablation was performed without preoperative biopsy (Figure 4). 

Histopathological and immunohistochemical examinations confirmed the diagnosis (Figure 5). 

No intraoperative complications, incidents or accidents were recorded. All patients enjoyed a fast and good quality recovery, without motor or sensory sequels, with maximum patient satisfaction.

### 3.3. Schwannoma

Schwannomas comprise about 5% of all benign soft tissue lesions, accounting for 27.5% of our study patients. Schwannomas can occur in people of any age and have no gender predisposition. Schwannomas usually manifest as solitary, slow-growing, encapsulated, painless lumps that persist long before diagnosis, leading to a more challenging treatment [17]. As the tumor grows and gradually compresses the nerve, pain, paresthesia, and other symptoms may appear. We report 22 cases of schwannomas with different anatomical sites and nerve involvement. 

Did not record any patients with type 2 neurofibromatosis association or multiple lesions. The following clinical signs supported the diagnosis: slow-growing tumor mass located along a peripheral nerve (large nerve trunk or even common or collateral digital nerves) of relatively hard consistency, sometimes painful spontaneously or especially on palpation, with a positive Tinel sign, mobile along the plane perpendicular to the nerve course, a suggestive sign of the presence of a schwannoma. We recorded a female/male ratio of 19:3 with an average age of 54.59 years. We identified eight cases on the digital collateral nerves of the long fingers, four on the digital collateral nerves of the thumb, six on the common digital nerves and four cases on the palmar median nerve. Imaging investigations consisted of two-view X-rays (to rule out a bone tumor), which revealed a minimal bone impression in two cases. Ultrasonography confirmed a nerve truck tumor in 11 of the 22 cases. In 14 cases, the MRI examination (which was not performed for all cases due to economic reasons) confirmed the diagnosis of schwannoma. Surgery was indicated in all cases and performed under local anesthesia (Walant) in patients with finger tumor localization and under loco-regional anesthesia (axillary block) for palmar lesions with median nerve involvement. Under the operating microscope, the tumor was enucleated, and the nerve fibers were kept intact to avoid postoperative neurological complications (sensitivity disorders) (Figure 6).

Histopathological examination confirmed the diagnosis of schwannoma, its characteristic features being detected: the presence of two areas of cellularity: Antoni A (compact hypercellularity) and Antoni B (myxoid hypocellularity). Immunohistochemical determinations were made for 15 out of 22 patients. In all cases, S100, CD 34 and collagen IV were positive, suggestive of the diagnosis of schwannoma; therefore, the diagnosis of peripheral malignant nerve trunk/sheath tumor was excluded (Figure 7).

The 2PD and SW tests were within normal range. No recurrence was recorded in any of the cases 2 years following surgery. Patient satisfaction was maximum in all cases, according to the MHQ scale.

### 3.4. Lipoma

Although lipoma is the most common form of benign soft tissue tumor, hand localization is rare, accounting for approximately 1% of the tumors in this region [4]. In our study group, out of the 80 patients included, 30 (37.5%) were diagnosed with lipoma, with a mean age of 53.06 years; 56.66% were male patients. We identified lipomas with different hand localizations; 13.3% of the cases were reported on the dorsal aspect of the hand, a very rare localization.

Development typically begins with an initial insidious growth period followed by a prolonged and latent maintenance state (between 1 and 8 years). Hand lipomas are often asymptomatic and only come to clinical attention once they grow large enough to induce mechanical impairment or if they are of cosmetic concern. (Figure 8).

The Posh sign was positive in all studied cases, with the limitation of thumb-digit tip pinch and reduced grasp in the cases with large palmar or thenar lipomas. Radiological examination revealed the integrity of the osteoligamentous structures and, through transparency, the presence of hand soft tissues tumor (Figure 9).

Ultrasonography was performed in all study cases and identified a well-defined homogeneous hyperechoic mass, thus confirming the diagnosis. MRI was performed on lipomas larger than 5 cm (13 cases) to detect possible malignancy signs. In all cases, complete lipoma ablation was performed. A histopathological examination confirmed the diagnosis. We did not identify any malignant component in any of our cases. The functional results were satisfactory in all cases, with full socio-professional reintegration and maximum patient satisfaction. No relapses were reported two years following surgery.

### 3.5. Epidermal Inclusion Cyst

Epidermal inclusion cysts are painless, benign, slow-growing soft tissue tumors that often occur months to years after a traumatic event [18]. We report three cases, males, with a mean age of 45 years, manual workers (in agriculture), with a prolonged and latent tumor maintenance state of 4 to 12 years. No patient reported any association between a history of trauma and the development of the tumor, nor did we detect the papillomavirus. The inclusion cysts in these patients were located in the palm and the volar aspect of the proximal phalanx of the middle finger. 

The patients presented to the clinic with a painless (spontaneously and on palpation) slow-growing tumor mass, relatively immobile on the deep planes. Sensory symptoms were absent. In one case, the large tumor size and the mid-palmar localization induced a reduced grip and pinch strength. The Posh sign was negative in all cases (differential diagnosis of lipoma). The radiological examination did not reveal osteoarticular changes or bone impressions but a translucent mass in the soft parts. MRI was performed for a precise diagnosis, showing the cystic nature of the mass and its intimate relations with the neighboring structures (Figure 10).

Surgery was performed under loco-regional anesthesia (axillary block). A wide surgical excision area was created through incisions parallel to palmar flexion creases (to avoid retractile scars), thus allowing the wide dissection and complete removal of the cysts (Figure 11).

Histopathological diagnosis of epidermal inclusion cyst diagnosed squamous epithelium and pericystic lymphoplasmacytic and basophilic infiltrate. The cyst encapsulated loose keratin material disposed of in a lamellar fashion (Figure 12).

Postoperatively, no motor or sensory sequels were recorded in the study patients, and they enjoyed full socio-professional reintegration.

### 3.6. Idiopathic Tenosynovitis with “Rice Bodies”

Tenosynovitis with “rice bodies” unrelated to rheumatic diseases (rheumatoid arthritis, systemic lupus erythematosus, seronegative arthritis, etc.) is rarely reported in the literature, especially in large series [19]. The present study reports eight cases of idiopathic tenosynovitis with “rice bodies”, accounting for 26.66% of the total 80 patients included in the study. 

Of these, 37.5% were female patients. The patients were investigated for rheumatic diseases, with negative results. In all cases, the mass was located on the volar aspect of the hand. X-ray examination reported no pathological findings. Ultrasonography described a well-defined small mass with mixed (liquid and solid) content in two cases in intimate contact with the flexors of the fourth and fifth fingers. The cystic nature of the mass was established, but further clinical findings suggestive of tenosynovitis with rice bodies are required to confirm the diagnosis; otherwise, an epidermal inclusion cyst is considered. MRI scan revealed heterogeneous masses with hyposignal on T2-weighted sequence and iso- and hypersignal on T1-weighted sequence. Inside, a multitude of tiny areas with hyposignal on T2-weighted sequence was described. Intraoperatively, we observed a pseudotumor with a relatively thick wall containing numerous formations in the form of yellow-white coins, “rice bodies” (Figure 13).

In all cases, the complete removal of the pseudotumors was performed under an axillary block (Figure 14).

We did not record any intraoperative incidents or immediate or late postoperative complications. Histopathological examination of the exeresis piece showed the presence of fibrin organized in the form of rice bodies, with an acidophilic amorphous center delimited by a thin fibrous layer. Aspects of proliferative synovitis with hyperplastic and hypertrophic synovial cells, as well as rich lymphoplasmacytic infiltrate, were also identified. Secretions were collected from the cyst content, and Ziehl–Nielsen stains were used to detect the presence of Koch bacilli or fungi, with negative results. Samples were collected to diagnose possible rheumatic or immune diseases or tuberculosis, with negative results, thus confirming the idiopathic character of the lesions. Postoperatively, from a functional point of view, the results were satisfactory, with full socio-professional reintegration. No recurrences were recorded within two years.

## 4. Discussion

Soft tissue tumors without skin involvement are relatively frequent in some anatomical regions but rare or very rare in the hand. Neurofibromas, benign tumors of nerve origin, account for approximately 5% of soft tissue tumors [10,12]. However, their location at the hand level is rare [14]. Nail bed location is extremely rare, with only 11 such cases being reported in the literature [10]. The clinical findings are nonspecific in these cases, and pathognomonic clinical signs are absent [14]. Good knowledge of the various types of hand tumors can avoid diagnostic errors or delays, as even MRI findings are nonspecific (impossible to differentiate from schwannoma). To confirm the diagnostic, histopathological and immunohistochemical examinations are required. Immunohistochemical examinations found positive S100 protein, with a much higher intensity in schwannomas. The treatment is surgical, consisting of complete tumor ablation [10,12,14]. The outcome of surgical treatment is excellent, without relapses, when the tumor is completely removed [10]. Although schwannoma is the most commonly diagnosed tumor of peripheral nerve origin, accounting for approximately 5% of all soft tissue tumors, its location in the hand is rare [20]. Our 22 cases reported in a series of 80 patients with soft tissue tumors make up the largest group reported up to now in the literature. The absence of specific clinical signs is characteristic of this type of tumor, but certain features can differentiate them from other tumors [2,21]. Tumor mobility in the transverse plane, with its significant limitation in the longitudinal plane, can guide the diagnosis [2]. Although the Tinel sign is present in 5% to 75% of schwannoma cases, in our study group, it was present in 100% [2,22]. The preoperative presence of neurological signs is suggestive of an unfavorable prognosis [2]. MRI examination can make the diagnosis of schwannoma in 90% [22,23]. However, no imaging study has a specificity above 95% in its diagnosis [23]. Not infrequently, its presence in the hand is not differentiated from lipoma or another tumor-like formation. Surgical enucleation of the tumor is the gold standard of treatment, thus preserving the nerve fibers and, at the same time, avoiding recurrence [4]. The present study did not record any postoperative neurological complications and recurrences within 2 years. The glomus tumor accounted for 1% to 4% of all hand tumors, with 65% of them located in the subungual region [16,24,25]. It can be solitary or multiple and is more commonly diagnosed in women, as the present study confirms [16,25]. Our report of 15 cases of glomus tumors of the hand in a group of 80 patients is also one of the largest cases reviewed in the literature. Unlike the other tumor types, glomus tumor is diagnosed with a hallmark symptomatic triad identified by Love’s pin test, Hildreth’s test, and cold sensitivity test [16]. The cold sensitivity test has a specificity of 100%, the most relevant and severe symptom of glomus tumors being intense pain on touch. However, cases of misdiagnosis have been reported [24,25]. MRI examination can be useful in making the diagnosis but has no relevant features for this type of tumor [26]. In the current study, MRI was not performed for economic reasons, and only a few patients could tolerate USG due to intense pain at the touch. Diagnostic certainty is obtained by immunohistochemistry reports positive for α-SMA, MSA and h-caldesmon in proportions of 99%, 95% and 87% of the cases. In glomus tumors, immunostaining for S100 is negative in the vast majority of cases [27]. Surgical outcomes are good, with high patient satisfaction and the absence of symptoms. Although lipoma is the most common benign tumor, finger lipomas are rare, with a reported incidence of 1% [4,28]. Of the 30 reported lipoma cases in our 80-patient study group, 17 were lipomas of the phalanx and long fingers, with males being most commonly affected; a traumatic event is invoked only in 9 cases. Just like the previous tumors, the clinical signs are not specific. The differential diagnosis includes any of the tumor categories detailed in the current study, as well as many other types of soft tissue tumors or tumor-like lesions [29,30]. The Posh sign can guide the clinical diagnosis, as it is always positive [4]. MRI examination has an essential role in diagnosing lipoma, with a predictability of 94%, and, most importantly, it can diagnose a possible malignant component of the tumor [4]. The recurrence rate of lipomas (5%) is strictly related to partial tumor removal [4,29]. In our study group, no relapse within two years was recorded. Epidermal inclusion cyst is a fairly common lesion, accounting for approximately 90% of cystic tumors [18,31]. Just like the other tumors included in the study, epidermal inclusion cyst is rarely found on glabrous skin and especially in the palm [18]. In the literature, relatively few cases of this type of injury are reported, especially of the hand [32]. Most frequently, the presence of an inclusion cyst is related to trauma or the presence of HPV [18]. In the three cases reported by us, none of these two were confirmed by patient history and laboratory tests. Being a painless, slow-growing tumor, it can reach relatively large sizes (4.5 cm in one of our study cases) when located in the palm. Lack of specific symptoms and pathognomonic clinical signs creates a challenging differential diagnosis, including diagnostics like lipoma, schwannoma, giant cell tumor, other types of cysts, etc. [32,33,34]. MRI is recommended to guide the diagnosis [18]. In our study, MRI was performed in only one of the cases with palmar location (for economic reasons); in the other cases, ultrasonography was the only imagistic examination performed [32]. Surgery aimed at the complete ablation of the cyst. Histopathology confirmed the diagnosis and identified stratified squamous epithelium in the cyst wall and large amounts of keratin [18]. In 1895 Reise was the first to describe “rice bodies” present in the joints of patients with tuberculosis [11,35]. Later, their presence was also associated with other diseases (seronegative arthritis, lupus erythematosus, rheumatoid arthritis, etc.), primarily affecting the bursae and joints [11,35]. The presence of tenosynovitis with “rice bodies” unrelated to any other rheumatic condition in an idiopathic context is rarely reported in the literature [11]. The literature reports include a series of reviews with 1 to 9 cases (Sugano—1 case, Forse—5 cases, Yamamoto—9 cases, Cegarra and Reda—1 case each, etc.) [11,35]. Our study reports a series of 8 cases of tenosynovitis with rice bodies. In the absence of specific symptoms, the differential diagnosis includes almost any type of tumor, especially in the absence of rheumatic diseases, tuberculosis, presence of HPV (which would guide the diagnosis toward an epidermal inclusion cyst) [11]. As in the case of any other type of tumor, neurological symptoms can appear when the tumor is enlarged. MRI examination can guide the diagnosis by identifying “rice bodies” [36,37]. Intraoperative findings are characteristic, describing a large mass of rice bodies in the context of tuberculosis or rheumatic diseases; idiopathic cases are very rare, and therefore, they are not being considered. We recorded rice body sizes ranging between 1.5 and 4 mm. Surgical treatment’s immediate and long-term results were good, requiring recovery through physiotherapy when the large-sized tumor-like lesion required large incisions and dissections. No relapse was reported at 2 years. Even after surgery, the patients were investigated in order to detect any associated disease that could explain the presence of “rice bodies”, but in all cases, no pathologic association was found [11]. 

## 5. Conclusions

Bringing together significant groups of rare tumors and tumor-like lesions of the hand, as well as highlighting their clinical and imaging features, in a single study greatly helps the physician to accurately assess the diagnosis—the definitive one being confirmed through the histopathological exam. Recognizing particular signs of these rare tumors may help avoid the iatrogenic lesions (e.g., knowing the mobilization direction characteristics of the schwannoma may be of use to avoid a biopsy with nerve fiber damage). The imaging exam has an important role in guiding the diagnosis, the most consequential being the MRI exam. The simple knowledge of the potential existence of the rare tumor-like lesion of the hand (e.g., an idiopathic tenosynovitis with “rice bodies” or a large epidermal inclusion cyst without traumatic or HPV infection context) may explain MRI aspects in a clinical context. The acknowledgement of the variety of rare tumors and tumor-like lesions of the hand is extremely important in establishing an early and accurate diagnosis, which leads to obtaining good therapeutic outcomes.

## Figures and Tables

**Figure 1 diagnostics-13-01204-f001:**
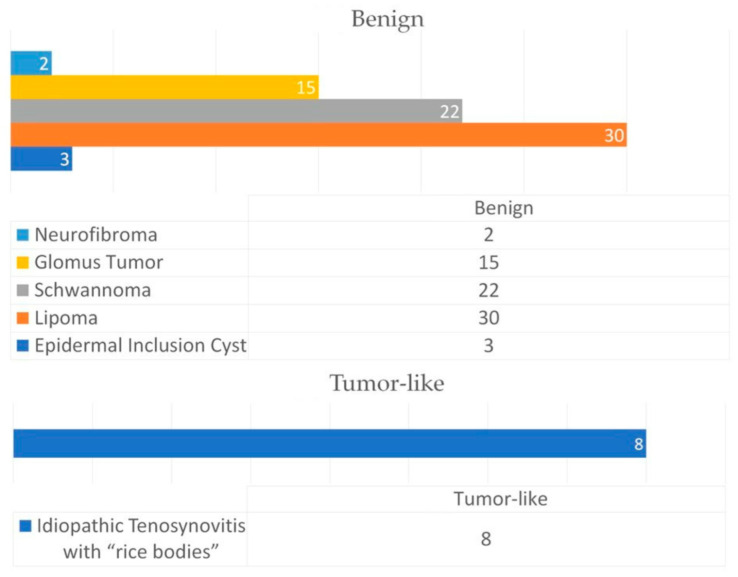
Types of soft tissue tumors diagnosed in the study group.

**Figure 2 diagnostics-13-01204-f002:**
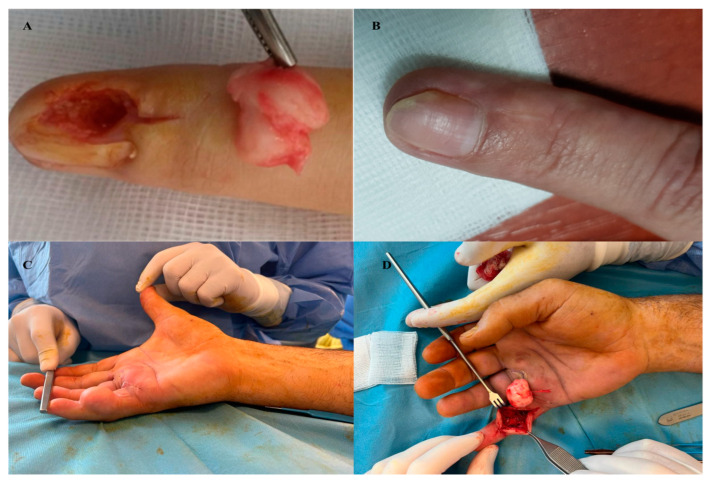
(**A**) intraoperative aspect, (**B**) Neurofibroma of the nail bed—1 year postoperatively result, (**C**) Large neurofibroma on the volar face of the fifth finger, right hand—preoperative aspect, (**D**) Large neurofibroma on the volar face of the fifth finger, right hand—tumor ablation and the recipient site.

**Figure 3 diagnostics-13-01204-f003:**
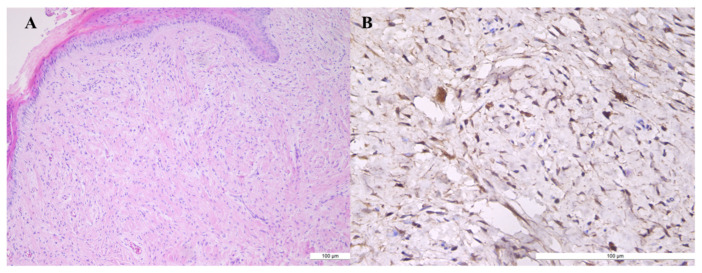
Neurofibroma (**A**) spindle cells with reduced cytoplasma; cells with ovoid, round or “comma” nulei; cells are separated by collagen fibers (HE × 10). (**B**) S100 protein staining positive in tumor cell.

**Figure 4 diagnostics-13-01204-f004:**
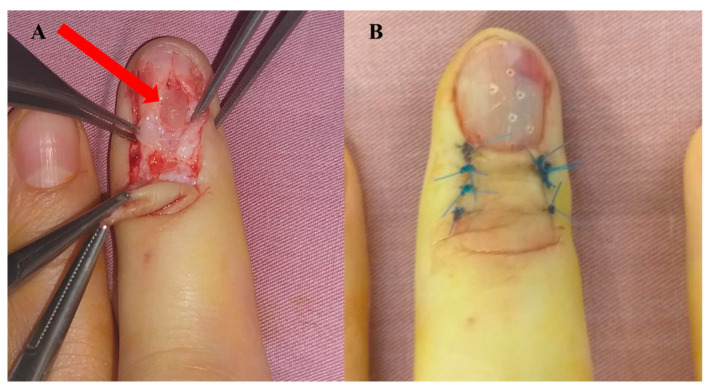
Glomus tumor of the middle finger nail bed. (**A**) Intraoperative appearance, (**B**) postoperatively result.

**Figure 5 diagnostics-13-01204-f005:**
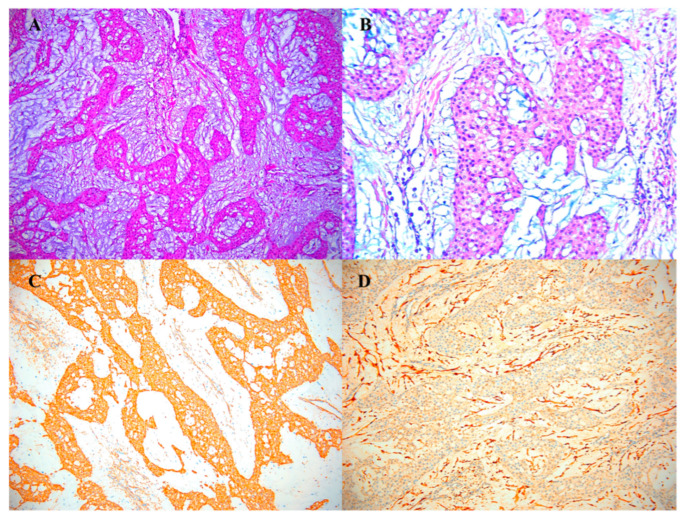
Glomus tumor. (**A**,**B**) Hematoxylin–Eosin (HE) glomic cells are small, uniform, often round, with a small, round nucleus placed centrally in the cell, the chromatin is homogeneous and the nucleoli are barely visible. The cytoplasm is amphophilic or pale eosinophilic. (**A**) HE × 2; (**B**) HE × 10; (**C**) positive marking for SMA; (**D**) negative marking for S100 positive.

**Figure 6 diagnostics-13-01204-f006:**
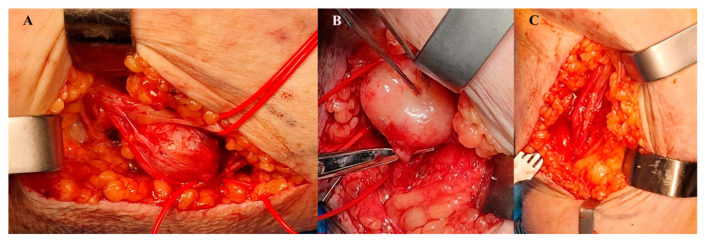
Schwannoma (**A**) intraoperative aspect, (**B**) intraoperative detail, (**C**) nerve after schwannoma ablation.

**Figure 7 diagnostics-13-01204-f007:**
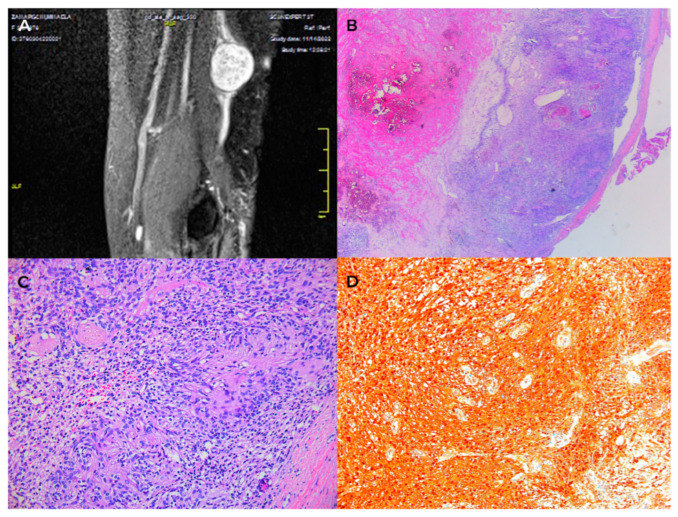
Schwanoma. (**A**) MRI aspect. (**B**,**C**) Encapsulated well-circumscribed lesion beneath the uninterrupted epidermis. The tumor is composed of different areas composed of different cellular densities. More cellular areas (Antoni A) are composed of a haphazard arrangement of bland cells with spindled and oval nuclei. Loose, less cellular areas (Antoni B) are composed of a loose oedematous and mucinous stroma with fibrillar collagen. The vessels are prominent and often surrounded by dense sclerosis. (**B**) HE × 10; (**C**) HE × 20; (**D**) Diffuse positive immunoreactivity for S100.

**Figure 8 diagnostics-13-01204-f008:**
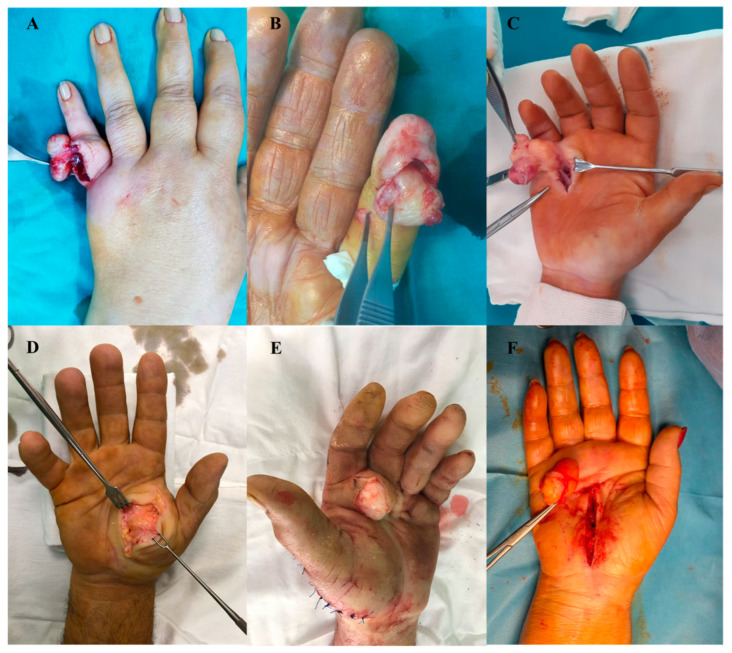
Uncommon sites of hand lipomas. (**A**) first phalanx of the little finger dorsal aspect. (**B**) pulp of the little finger. (**C**) ulnar palmar region. (**D**) thenar eminence. (**E**) radial palmar region, (**F**) medioplamar region.

**Figure 9 diagnostics-13-01204-f009:**
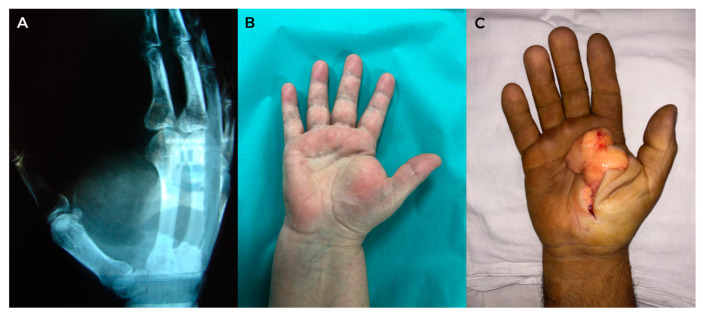
Lipoma (**A**) radiological appearance of hand lipoma—homogeneous soft tissue lucency between the 1st and 2nd metacarpal bones; (**B**) thenar lipoma—preoperative view; (**C**) thenar lipoma—intraoperative view.

**Figure 10 diagnostics-13-01204-f010:**
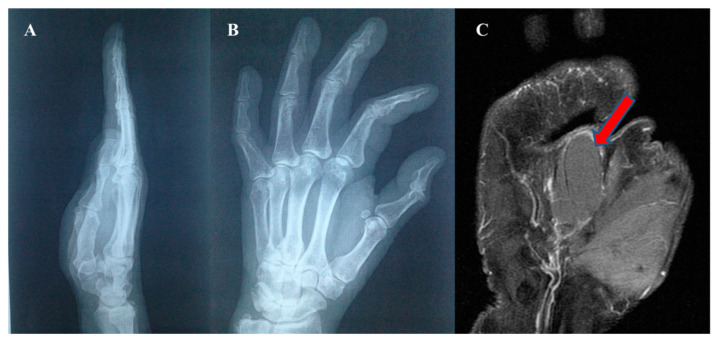
Epidermal inclusion cyst of the palm. (**A**,**B**) radiological aspect, (**C**) MRI aspect.

**Figure 11 diagnostics-13-01204-f011:**
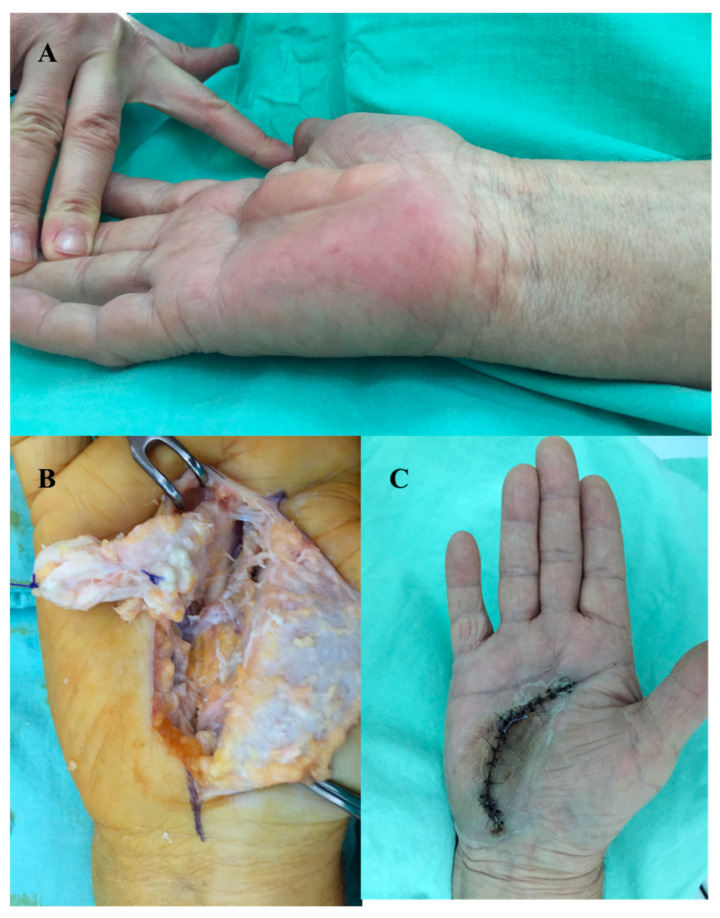
Epidermal inclusion cyst (**A**) preoperative aspect, (**B**) intraoperative aspect, (**C**) postoperative aspect (ten days after surgery).

**Figure 12 diagnostics-13-01204-f012:**
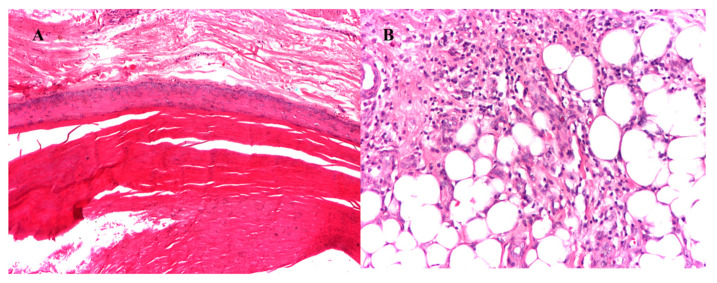
Epidermal inclusion cyst. (**A**) cystic wall—granular layer with lamellated keratine (HE × 10), (**B**) inflammatory infiltrated—adipose tissue (HE × 20).

**Figure 13 diagnostics-13-01204-f013:**
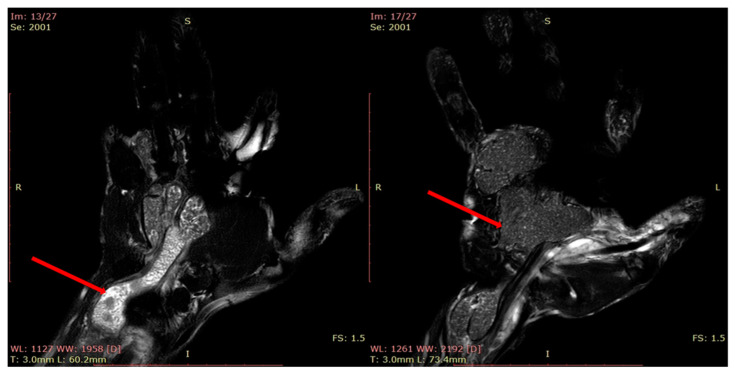
Idiopathic tenosynovitis with “rice bodies”—MRI aspect.

**Figure 14 diagnostics-13-01204-f014:**
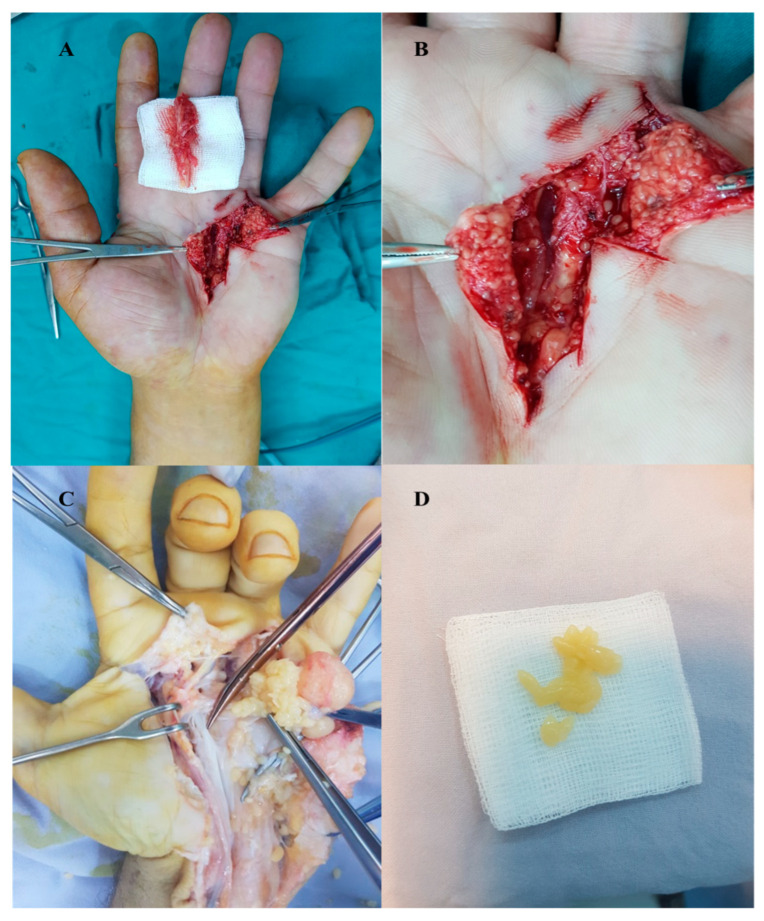
Tenosynovitis with “rice bodies” (**A**,**B**) localization on the ulnar aspect of the palm (**B**) detail), (**C**) mediopalmar localization, (**D**) “rice bodies”.

**Table 1 diagnostics-13-01204-t001:** Epidemiological factors and history of the tumors in the study group.

	Diagnostic	Cases	F	M	Age (Mean)	History
1	Neurofibroma	2	2	0	34 y	9 m–1 y
2	Glomus Tumor	15	13	2	40.13 y	6 m–3 y
3	Schwanomma	22	19	3	54.49 y	1–3 y
4	Lipoma	30	13	17	53.06 y	1–8 y
5	Inclusion Epidermal Cyst	3	0	3	45 y	4–12 y
6	Idiopathic Tenosynovitis with “rice bodies”	8	3	5	54.87 y	1–4 y

F = female, M = male, m = month, y = year, m = month.

**Table 2 diagnostics-13-01204-t002:** Clinical and macroscopic aspects of the tumors in the study group.

	Diagnostic	Clinical Findings	Clinical Tests	Macroscopic Intraoperative Aspects
1	Neurofibroma	- Painless tumor - Significant onychodystrophy in cases with nail bed involvement - Unsightly appearance	- Tinel sign +	Solid, well-defined nodule-like tumor. White-pink aspect in cases with nail bed involvement
2	Glomus Tumor	- Pain of high intensity to touch, pressure - Modifications in the nail bed or slight situs elevation (depending on location)	- Love’s test + - Cold sensitivity test + - Hildreth’s test + - Transilumination test +	Nodular lesion of small dimensions, well defined, with translucent or white wall
3	Schwanomma	- Paresthesias - Pain, numbness and fatigue may take place with the increasing size of the tumor.	- Tinel sign + - 2PD - Transversal mobility - Relatively immobile in the longitudinal plane	Consistent tumor mass, well defined, mobile in the transverse plane, translucent white
4	Lipoma	- Pain and paresthesias - Finger flexion/extension - impairment (mid-palm lipoma) - Carpal tunnel syndrome symptoms (wrist lipoma) - Functional impotence caused by large tumor size - Unsightly appearance	- Posh test + - Pinch test - 2PD test - pinch movements, grasping - fingers mobility	Relatively mobile tumor mass with a soft consistency was detected in cases with wrist, thenar eminence and finger location. Immobile tumor mass in cases with mid-palm localization
5	Inclusion Epidermal Cyst	- Painless tumor - Functional impotence caused by large tumor size - Unsightly appearance	- Posh test—(for differential diagnosis with lipoma) - 2PD - pinch test - grasping - fingers mobility	Relatively firm, fixed on deep planes, covered by teguments with normal appearance. Thin white capsule, relatively friable, with yogurt-like content, in some cases with intimate contact with other structures (tendons, vascular-nerve bundles)
6	Idiopathic Tenosynovitis with “rice bodies”	- Painless swelling mass - Painful/Paresthesias depending on the size of the tumor	- Posh sign—(for differential diagnosis with lipoma) - 2PD - pinch test - grasping - fingers mobility	

**Table 3 diagnostics-13-01204-t003:** The findings of the imaging investigations of the tumors in the study group.

	Diagnostic	Rx Exam	USG	MRI
1	Neurofibroma	- Absence of skeletal changes - There may be bone impressions	Nodular tumor mass, well delimited, with polycyclic appearance, hypoechoic	-
2	Glomus Tumor	- Absence of skeletal changes - There may be bone impressions	Difficult to examine due to the pain produced by simple touch. Small, well-defined, hypoechoic tumor. Echo-doppler—intratumoral vascularization	-
3	Schwanomma		The tumor mass is well-defined, hypoechoic, homogeneous, in direct connection with the peripheral nerve from which it originates	Tumor mass with moderate hyperintense feature in T1-weighted images and hyperintense feature on fluid-sensitive aspects
4	Lipoma	- Absence of skeletal changes - Tumor mass visualized through radio-transparency	Well-defined hyperechoic, homogeneous tumor mass. Vascularization: absent	Well-defined tumor mass within its own capsule, with lipomatous signal (hypersignal in T1 and T2 and hyposignal in STIR images). In some cases—fine septa inside the tumor with discrete gadolinophilia
5	Inclusion Epidermal Cyst	- Absence of skeletal changes - Tumor mass visualized through radio-transparency	Well-defined tumor mass with a relatively thick wall and homogeneous content	Well-defined cystic tumor formation, with hypersignal in T2 and hyposignal in T1
6	Idiopathic Tenosynovitis with “rice bodies”	- Absence of skeletal changes - Tumor mass located in the soft parts (visualized through transparency) in the case of larger ones	Well-defined tumor mass with both solid and liquid content (mixed)	Well-defined lobular, multiloculated tumor formation, heterogeneous with hyposignal in T1 and iso/hypersignal in T2. Numerous small hyposignal areas in T2; T1-weighted sequence showing a mass with low signal intensity surrounding the tendons

Rx exam = radiological exam, USG = ultrasonography, MRI = magnetic resonance imaging.

**Table 4 diagnostics-13-01204-t004:** Histopathological and immunohistochemical features of the tumors in the study group.

	**Diagnostic**	**HP**	**IHC**	
1	Neurofibroma	Non-incapsulated, low and moderate cellularity, cells loosely arranged; spindle shape cells with small amount of cytoplasm; cells with smaller size than schwannoma cells; cells with round, ovoid or comma-shaped nuclei separated by collagen fibers and myxoidmateria, lnerve not often identified especially in solitary tumors.	S100	+
			CD34	+
2	Glomus Tumor	Insular and trabecular architecture; the presence of a fibrous capsule at the periphery; small, uniform, well-demarcated glomerular cells; small round nuclei centrally located in the cell; homogeneous chromatin; pale nucleoli; amphophilic or pale eosinophilic cytoplasm.	α-SMA	+
			CD34	+
			h-Caldesmon	+
			S100	−
			CK AE1/AE3	−
			P63	−
3	Schwanomma	Encapsulated biphasic tumors with compact areas of spindle cells (Antoni A) alternating with loose foci (Antoni B); some hypercellularity; infrequent collagen bundles; Verocay bodies showing palisade; nerves often identifiable; o NF1, occasionally NF2. Cystic degeneration can sometimes be visualized.	S100	+
			CD34	+
			Collagen IV	+
			SMA	−
4	Lipoma	Mature adipose tissue; the fat contains small capillaries within thin fibrous strands; thin fibrous capsule is often seen; necrosis and other inflammatory changes may be seen when lipomas are traumatized.	-	
5	Inclusion Epidermal Cyst	Cyst lining by stratified squamous epithelium with a granular layer—key feature. No significant nuclear atypia. Contains keratin–acellular, lamellar appearance.+/− Granulomatous inflammation due to rupture.	-	
6	Idiopathic Tenosynovitis with “rice bodies”	Proliferative synovitis with synovial cell hyperplasia and hypertrophy; Fibrin organized in the form of rice bodies, some nodules with central cartilaginous transformation; Fibrotic nodules surrounded by histiocytes; Epithelioid granulomas with multinucleated giant cell, some of them Langhans cell-like.	-	

HP = histopathology, IMC = immunohistochemistry.

## Data Availability

The data can be obtain from the corresponding author upon resonal request.

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
