# Peer review of "Rare Benign Tumors and Tumor-like Lesions of the Hand without Skin Damage—Clinical, Imagistic and Histopathological Diagnosis, Retrospective Study"

_diagnostics, 2023, doi:10.3390/diagnostics13061204_

Round 1

Reviewer 1 Report

A scientific work to be reviewed must have the lines enumerated to allow the reviewer to indicate where a correction should be made. Several tumors of rare presentation in the region of the hands are well described. However, no useful message to the clinician can be found in the conclusions of the work, despite is written that “a knowledge of hand tumors types and their clinical and laboratory characteristics is necessary for every surgeon”.

I think it is necessary:

·       Enumerate the lines

·       Elaborate a final a message giving more information than a description of the various tumors

·       indicate and emphasize which clinical and laboratory examinations can be decisive in distinguishing pathology

subchapter 3.1 needs more precise referrals, such as : doi: 10.1111/dth.15355.

Actually in my opinion the paper is not yet ready to be published. It will after response to queries.

Author Response

Dear Reviewer,

Thank you very much for revising the manuscript diagnostics-2262347and for comments made!

According to the recommendations:

  • I enumerated the lines
  • I elaborated a final message giving more information regarding the importance and conclusions of the study.
  • I indicated the importance of the clinical and imagistic examinations which can be decisive (in conclusions).
  • In subchapter 3.1 I have included more precise references as doi: 10.1111/dth.15355 (reference no. 15)

Thank you,

Best regards,

Mihaela Pertea MD PhD

Author of the manuscript diagnostics-2262347

Reviewer 2 Report

This is a retrospective data collection of the authors' experience on benign hand tumors. The authors presented cases with neurofibromas, glomus tumors, schwannomas and other benign tumors. However, most of the tumors presented in this paper are well documented in the textbooks, and thus, this paper lacks novelty. I am not sure whether the paper is of interest for readers. Another concern is redundant figures. Fig. 2, 5, 8, 11, 14 and 18 have little information and should be deleted.

Author Response

Dear Reviewer,

Thank you very much for revising the manuscript diagnostics-2262347 and for comments made!

According to the recommendations:

  • I improved the conclusions, highlighting the importance of the study
  • I deleted figures mentioned : 2,5,8,11,14,18, that really didnt’t add anything.

Thank you very much!

Best regards,

Mihaela Pertea MD PhD

Author of the manuscript diagnostics-2262347

Round 2

Reviewer 2 Report

Thank you for the revised version. I have no additional comment.

Author Response

Dear reviewer,

Thank you for the reviewing the manuscript, for recommendations and appreciations!

Best wishes,

Mihaela Pertea MD PhD
